# Rate enhancement in collisions of sulfuric acid molecules due to long-range intermolecular forces

**Roope Halonen[1], Evgeni Zapadinsky[1], Theo Kurtén[2], Hanna Vehkamäki[1], and Bernhard Reischl[1]**

[1]Institute for Atmospheric and Earth System Research / Physics, Faculty of Science, University of Helsinki, P.O. Box 64, FI-00014, Finland

[2]Institute for Atmospheric and Earth System Research / Chemistry, Faculty of Science, University of Helsinki, P.O. Box 55, FI-00014, Finland

**Correspondence:** Roope Halonen (roope.halonen@helsinki.fi)

**Abstract.** Collisions of molecules and clusters play a key role in determining the rate of atmospheric new particle formation and growth. Traditionally the statistics of these collisions are taken from kinetic gas theory assuming spherical non-interacting particles, which may significantly underestimate the collision coefficients for most atmospherically relevant molecules. Such systematic errors in predicted new particle formation rates will also affect large-scale climate models. We have studied the statistics of collisions of sulfuric acid molecules in vacuum by atomistic molecular dynamics simulations. We have found that the effective collision cross section of the $H_2SO_4$ molecule, as described by an OPLS-All Atom force field, is significantly larger than the hard-sphere diameter assigned to the molecule based on the liquid density of sulfuric acid. As a consequence, the actual collision coefficient is enhanced by a factor 2.2 at 300 K, compared to kinetic gas theory. This enhancement factor obtained from atomistic simulation is consistent with the discrepancy observed between experimental formation rates of clusters containing sulfuric acid and calculated formation rates using hard sphere kinetics. We find reasonable agreement with an enhancement factor calculated from the Langevin model of capture, based on the attractive part of the atomistic intermolecular potential of mean force.

## 1   Introduction

New particle formation from condensable vapours gives an important contribution to the composition of aerosols in the atmosphere which affects air quality as well as the Earth's climate. The positive and negative contributions of atmo-spheric aerosols to the planet's radiative balance are still not fully understood, and currently constitute one of the largest uncertainties in climate modelling. The earliest stage of new particle formation involves collisions of individual molecules leading to the appearance of a new molecular complex. In many theoretical approaches, the statistics of such collisions are simply taken from kinetic gas theory, i.e. the molecules are considered to be non-interacting hard spheres, and a collision occurs when the impact parameter i.e. the perpendicular distance between the spheres' trajectories is smaller than the sum of the hard spheres' radii. The hard sphere collision cross section is independent of the relative velocity of the colliding bodies, and the collision rate coefficient for hard spheres of identical radii is customarily expressed as

$$\beta_{HS} = \sqrt{\frac{8k_BT}{\pi\mu}}\pi(2R)^2, \tag{1}$$

where $k_B$ is Boltzmann's constant, $T$ is the temperature, $\mu$ is the reduced mass and $R$ is the radius of the spheres.

It is well known that acid-base clusters, and in particular clusters containing sulfuric acid and ammonia, or amines, are very relevant in nucleation and growth of particles that can serve as cloud condensation nuclei (Almeida et al., 2013). Such molecules, however, are not necessarily spherical and despite being charge neutral, exhibit long-ranged attraction due to interactions between permanent dipoles, permanent and induced dipoles, or induced dipoles (Israelachvili, 2011). Therefore, it is reasonable to expect that particle growth rates or cluster size distributions predicted using collision coefficients from kinetic gas theory will have a systematic error, which needs to be accounted for. In fact, systematic discrepancies have been found between experimental particle forma-

tion rates and values predicted from kinetic modelling and cluster dynamics simulations, where hard-sphere collisions are assumed. Kürten et al. (2014) measured the kinetic formation rate of sulphuric acid dimers and found that an enhancement factor of 2.3 needed to be applied to the formation rate obtained from a kinetic model. Lehtipalo et al. (2016) and Kürten et al. (2018) have studied particle formation rates in systems containing sulphuric acid, dimethylamine and water and concluded that an enhancement factor of 2.7 and 2.3, respectively, was needed to match experimental particle formation rates.

The effect of long-range interactions between neutral polar molecules on the capture rate constant has been studied by classical trajectory integration (Maergoiz et al., 1996c). The interaction potential between the colliding parties has been approximated by two terms: First, an anisotropic interaction between permanent dipoles, proportional to $r^{-3}$, where $r$ is the distance between the centres of mass of the molecules. Second, an isotropic term due to the interaction between permanent dipole and induce dipole, and the interaction between induced dipoles, proportional to $r^{-6}$. However, such an approximation is inaccurate when the distance between the colliding particles is comparable to their size. Rate coefficients for ion-molecule capture processes have also been studied theoretically in both classic and quantum regime (Moran and Hamill, 1963; Su and Bowers, 1973; Su et al., 1978; Chesnavich et al., 1980; Clary, 1985; Troe, 1987) or by using trajectory calculations (Dugan Jr. and Magee, 1967; Chesnavich et al., 1980; Su and Chesnavich, 1982; Maergoiz et al., 1996a, b). Atomistic simulations have been used to study collisions of Lennard-Jones clusters and atmospherically relevant molecules, but these studies did not analyze or report thermal collision rate coefficients (Napari et al., 2004; Loukonen et al., 2014). Recently, Yang et al. (2018) have studied the condensation rate coefficients for Au and Mg clusters at various gas temperatures using molecular dynamics calculations. The influence of Van der Waals forces on the collision rate has also been considered in Brownian coagulation models of ultra-fine aerosol particles (Marlow, 1980; Sceats, 1986, 1989).

In the present work, we use atomistic molecular dynamics simulations to study the statistics of collisions between sulfuric acid molecules in vacuum, determine the collision rate coefficient and calculate the enhancement factor over kinetic gas theory. We are here focusing on "reactive" collisions, defined by the formation of one or more hydrogen bonds between the molecules. Detailed modelling of e.g. proton transfer processes related to hydrogen bond formation in such reactive collisions would require first principle simulations (Loukonen et al., 2014), however the need to simulate a large number of individual trajectories to cover a representative range of impact parameters and relative velocities makes this impossible. In the present study we are modelling the collision rate enhancement due to long-range interactions, which can be decently described by empirical force fields.

In section 2 we discuss technical matters of the choice of force field and the simulation setup and give a brief overview of the theoretical background for collisions of atmospheric particles. In section 3, simulation results are presented, discussed within the theoretical framework and compared to analytical and experimental results.

## 2   Simulation details and theoretical models

### 2.1   Force field benchmark

We have considered two force fields to describe the sulfuric acid molecules in the present study. The first choice was the force field by Ding et al. (2003), fitted specifically to reproduce DFT structures and energies of small clusters of sulfuric acid, bisulfate and water, in vacuum. The second choice was the force field by Loukonen et al. (2010), who had fitted interaction parameters for sulfuric acid, bisulfate and dimethylammonium according to the OPLS-All Atom procedure (Jorgensen et al., 1996). Both force fields are fitted to reproduce the $C_2$ geometry of the isolated $H_2SO_4$ molecule in vacuum, and the atoms' partial charges create dipole moments of 3.52 and 3.07 Debye, for Ding et al. and Loukonen et al., respectively, in agreement with experiments (2.7–3.0 Debye) and ab initio calculations (2.7–3.5 Debye) (Sedo et al., 2008). In both force fields intermolecular interactions are described by the sum of Lennard-Jones potentials between atoms $i$ and $j$ separated by a distance $r_{ij}$, with distance and energy parameters $\sigma_{ij}$ and $\varepsilon_{ij}$, and Coulomb interactions between the partial charges $q_i$ and $q_j$,

$$
\begin{aligned}
U_{\text{inter}} \;=\;& \sum_{i=1}^{N_1}\sum_{j=1}^{N_2} 4\varepsilon_{ij}\left[\left(\frac{\sigma_{ij}}{r_{ij}}\right)^{12} - \left(\frac{\sigma_{ij}}{r_{ij}}\right)^{6}\right] \\
& + \sum_{i=1}^{N_1}\sum_{j=1}^{N_2}\frac{1}{4\pi\epsilon_0}\frac{q_i q_j}{r_{ij}}.
\end{aligned} \tag{2}
$$

However, in the force field by Ding et al., the geometry of the individual molecule is simply constrained by harmonic potentials with force constants $k_{ij}$ between all pairs of atoms,

$$
U_{\text{intra}}^{\text{Ding}} = \sum_{i=1}^{N_1-1}\sum_{j=i+1}^{N_1} \frac{k_{ij}}{2}\left(r_{ij} - r_{ij}^0\right)^2, \tag{3}
$$

while in OPLS the intramolecular interactions consist of the usual sum of two, three, and four-body potentials, i.e. harmonic bonds between covalently bonded atoms, harmonic angles $\theta$ between atoms separated by two covalent bonds, and torsions (dihedral angles $\phi$) between atoms separated by

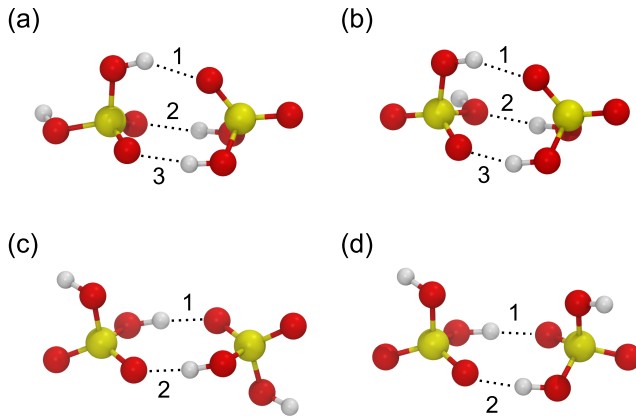

**Figure 1.** Four minimum energy structures for the sulfuric acid dimer (a–d) used to benchmark the force fields by Ding et al. (2003) and Loukonen et al. (2010) against ab initio calculations by Temelso et al. (2012). Sulfur atoms are yellow, oxygens red and hydrogens white. Hydrogen bonds are indicated by dotted lines and enumerated according to Tab. 1.

three covalent bonds,

$$U_{\mathrm{intra}}^{\mathrm{OPLS}} = \sum_{i=1}^{N_{\mathrm{bonds}}} \frac{k_i^b}{2} \left(r_i - r_i^0\right)^2 + \sum_{j=1}^{N_{\mathrm{angles}}} \frac{k_j^\theta}{2} \left(\theta_j - \theta_j^0\right)^2$$
$$+ \sum_{k=1}^{N_{\mathrm{dihedrals}}} \sum_{n=1}^{4} \frac{V_n}{2} \left[1 + \cos(n\phi^k - \phi_n^k)\right]. \qquad (4)$$

To validate the force fields, we compare the structures and energies of four stable configurations of the sulfuric acid dimer illustrated in Fig. 1(a–d) to ab initio structures and energies at the RI-MP2/CBS//6-31+G* level of theory calculated by Temelso et al. (2012). The results of the benchmark are summarised in Tab. 1. The force field by Ding et al. correctly predicts the lowest energy for dimer structure "a", and the relative energy differences $\Delta\Delta E$ between optimised structures are closer to those obtained in the ab initio calculation than for the OPLS force field, which assigns the lowest energy to structure "d", which is the highest energy structure in the ab initio calculation. The geometries of the structures agree well with the ab initio result for both force fields, with the OPLS force field reproducing the ab initio hydrogen bond lengths slightly better than the force field by Ding et al. The binding energies at $T = 0$ K are slightly lower for the force fields ($-0.64$ and $-0.67$ eV for OPLS, and Ding et al., respectively) compared to the ab initio value of $-0.72$ eV. Overall, the force field by Ding et al. performs slightly better in terms of energetics.

However, the vibrational spectra, calculated from the Fourier transform of the velocity autocorrelation functions of an isolated $H_2SO_4$ molecule in vacuum, exhibit strong differences: while the force field by Loukonen et al. is able to reproduce the experimental and ab initio spectra very well

(Hintze et al., 2003; Chackalackal and Stafford, 1966; Miller et al., 2005), the force field by Ding et al. is not, as shown in Fig. 2. Intramolecular vibrations are relevant in the context of molecular collisions, e.g. when studying energy transfer between different internal degrees of freedom during, and after the collision. Also, the OPLS-All Atom procedure allows for transferable potentials, as opposed to the Ding et al. force field which cannot easily be extended to other chemical compounds in future studies. For these two reasons, we decided to use the OPLS force field by Loukonen et al. for the collision simulations.

## 2.2 Potential of mean force of two sulfuric acid molecules

We first calculated the binding free energy of two sulfuric acid molecules in vacuum as described by the force fields of Loukonen et al. and Ding et al. The potential of mean force (PMF) as a function of the sulfur–sulfur distance was calculated from a well-tempered metadynamics simulation (Barducci et al., 2008), using the PLUMED plug-in (Tribello et al., 2014) for LAMMPS (Plimpton, 1995). We used a Velocity Verlet integrator with a time step of 1 fs, to correctly resolve the motion of the hydrogen atoms. The Lennard-Jones interactions were cut off at 14 Å and electrostatic interactions were only evaluated in direct space, with a cut-off at 40 Å. We employed 24 random walkers and Gaussians with a width of 0.1 Å and initial height of $k_BT$ were deposited every 500 steps along the collective variable and a harmonic wall was used to restrict it to values below 35 Å. A bias factor of 5 was chosen, and a Nosé–Hoover chain thermostat of

**Table 1.** Relative energies $\Delta\Delta E$ (eV) and hydrogen bond distances $d_{\mathrm{O\cdots H}}$ (Å) for the sulfuric acid dimer structures (a–d) in Fig. 1 obtained from ab initio calculations by Temelso et al. (2012) and with the force fields by Ding et al. (2003) and Loukonen et al. (2010), following the OPLS doctrine.

| Structure | | | a | b | c | d |
|---|---|---|---|---|---|---|
| $\Delta\Delta E$ | ab initio | | 0.000 | 0.032 | 0.036 | 0.048 |
| | Ding et al. | | 0.000 | 0.081 | 0.052 | 0.045 |
| | OPLS | | 0.180 | 0.099 | 0.004 | 0.000 |
| $d_{\mathrm{O\cdots H}}$ | ab initio | 1 | 1.82 | 1.74 | 1.75 | 1.75 |
| | | 2 | 1.89 | 1.91 | 1.75 | 1.75 |
| | | 3 | 1.90 | 1.87 | | |
| | Ding et al. | 1 | 2.00 | 1.84 | 1.75 | 1.74 |
| | | 2 | 1.87 | 2.31 | 1.74 | 1.74 |
| | | 3 | 1.88 | 1.85 | | |
| | OPLS | 1 | 1.91 | 1.84 | 1.72 | 1.72 |
| | | 2 | 1.85 | 1.87 | 1.72 | 1.72 |
| | | 3 | 1.83 | 1.83 | | |

Energy unit conversion: $1$ eV $\approx 96.49$ kJ·mol$^{-1}$ $\approx 23.06$ kcal·mol$^{-1}$ $\approx 38.68 k_BT$ at $T = 300$ K.

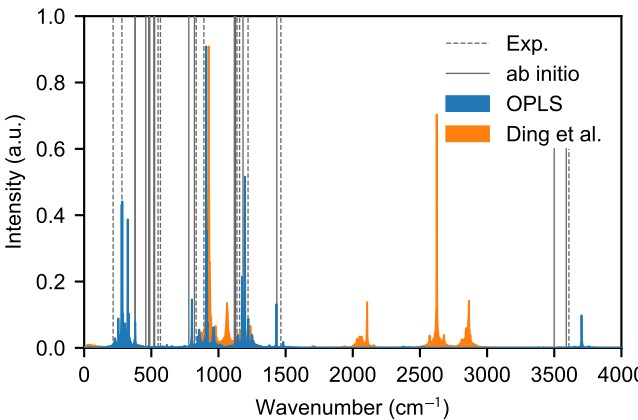

**Figure 2.** Vibrational spectra of the sulfuric acid molecule obtained with OPLS (Loukonen et al., 2010) (blue area) and the force field by Ding et al. (2003) (orange area). The spectra of Ding et al. has its highest frequency mode at 6565 cm$^{-1}$ outside of the range of the figure. The positions of the peaks in the experimental spectra (Hintze et al., 2003; Chackalackal and Stafford, 1966; Miller et al., 2005) and ab initio calculations (Miller et al., 2005) are indicated by dashed and solid grey lines, respectively.

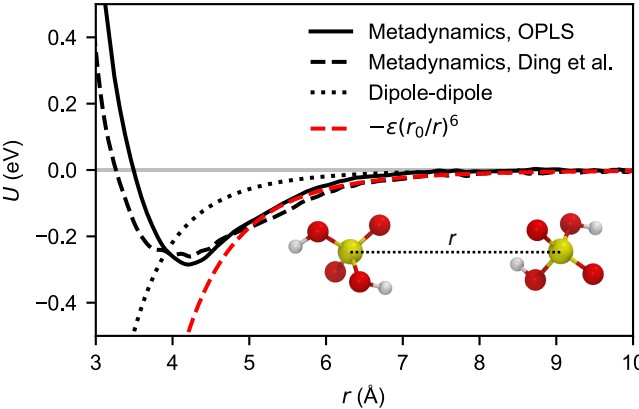

**Figure 3.** Potential of mean force between two $H_2SO_4$ molecules as a function of the sulfur–sulfur distance calculated by metadynamics simulation for the OPLS force field (Loukonen et al., 2010) (solid black line) and the force field by Ding et al. (2003) (dashed black line) at 300 K. For comparison, the Keesom (thermally averaged permanent dipole–permanent dipole) interaction between two point dipoles (see Eq. (A11)) of 3.07 Debye is depicted by the dotted line. The dashed red line shows the attractive potential (Eq. (8)) based on the calculated PMF curve of the OPLS force field.

length 5 with a time constant of 0.1 ps was used to keep a temperature $T = 300$ K. The combined length of the trajectories was 120 ns for each force field. Both PMFs, shown in Fig. 3, exhibit a minimum at $r = 4.1$ Å, and the binding free energies are $\Delta F = -0.29$ eV and $-0.27$ eV for Loukonen et al., and Ding et al., respectively. This is in excellent agreement with the ab initio value of $\Delta G = -0.30$ eV obtained from Boltzmann-averaging over the four minimum energy dimer structures by Temelso et al. (2012) at 298.15 K, and more recent calculations at higher level of theory, which predict slightly weaker binding ($-0.23$ to $-0.26$ eV) (Elm et al., 2016; Myllys et al., 2017).

## 2.3 Collision simulation setup

Molecular dynamics simulations were performed with the LAMMPS code, using a Velocity Verlet integrator with a time step of 1 fs. The Lennard-Jones interactions were cut off at 14 Å and electrostatic interactions were only evaluated in direct space, with a cut-off at 120 Å. The simulations were carried out in the NVE ensemble, as the colliding molecules constitute a closed system (in atmospheric conditions collisions with the carrier gas are rare on the time scale of collisions between sulfuric acid molecules). In order to determine the molecules' collision probability as a function of impact parameter and relative velocity, the following setup was used: first, two sulfuric acid molecules were placed in the simulation box, separated by 100 Å along x and the impact parameter $b$ was set along the z direction, ranging from 0 to 17.5 Å in steps of 0.5 Å. Atomic velocities were randomly assigned from a Maxwell–Boltzmann distribution at $T = 300$ K, and

the centre of mass motion of each molecule removed separately. Then the system was evolved for 50 ps, to randomise the intermolecular orientation and ensure equipartition of energy along the intramolecular degrees of freedom. At $t = 50$ ps, each molecule received a translational velocity along the x direction, $v_x = \pm v/2$, were $v$ denotes the relative velocity, to set them on a potential collision course. The simulation was continued for another 250 ps, to ensure the possibility of a collision, even at the smallest relative velocities. For a collision of two identical molecules with molecular mass $m$, the relative velocities follow the Maxwell–Boltzmann distribution with reduced mass $\mu = m/2$. We sampled relative velocities between 50 and 800 ms$^{-1}$, in steps of 50 ms$^{-1}$. 99 % of the distribution lies within this range at $T = 300$ K. For each value of the impact parameter $b$ and the relative velocity $v$, 1000 simulations were carried out starting with different initial atomistic velocities, to ensure good sampling. The simulation setup is very similar to the one recently used by Yang et al. (2018). Sulfuric acid molecules bind to each other through the formation of one or more hydrogen bonds. However, even if a collision course leads to an attachment of the two molecules, a portion of the kinetic energy will be redistributed on the degrees of freedom of the formed complex, and this excess energy can lead to a rapid dissociation in the absence of a thermalizing medium. To automate the analysis of over half a million individual trajectories, we define a collision as a trajectory during which the electrostatic energy ($E_{\mathrm{Coul}}$) is lower than a threshold value of $-0.25$ eV in at least 10 consequent frames (100 fs), indicative of the

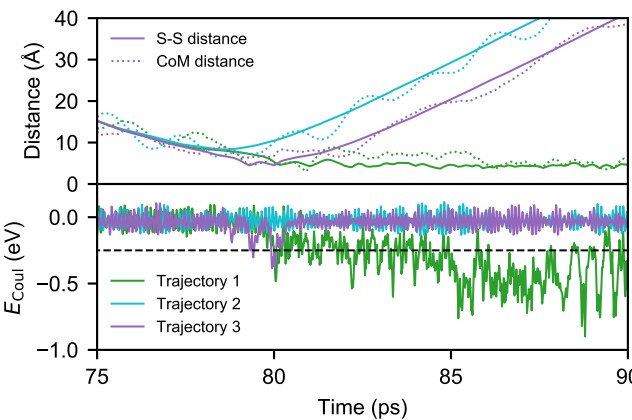

**Figure 4.** Three example MD trajectories with relative velocity of $350 \text{ ms}^{-1}$ and impact parameter of 8.5 Å after 50 ps of energy relaxation and first 25 ps of collision simulation. The upper panel shows both the distance between the two sulfur atoms (solid lines) and the centre-of-mass distance of the molecules (dotted lines) during the simulations. The electrostatic energies $E_{\text{Coul}}$, which is used to determine the possible collision event, for successful (1), unsuccessful (2) and successful but instantly evaporating (3, continuously 0.18 ps below the threshold) trajectories are shown in the lower panel with the threshold value (dashed black line).

formation of one or more hydrogen bonds. Three examples of simulated trajectories with relative velocity closest to the mean velocity at 300 K ($350 \text{ ms}^{-1}$) and impact parameter of 8.5 Å are illustrated in Fig. 4.

The results from the atomistic simulations will be compared to different theoretical models described in the following.

## 2.4  Classical model of capture in a field of force

As the collision rate in the context of atomistic simulations is
defined as the reaction rate of hydrogen bonding, the related theoretical models are often based on the assumption that if the trajectory of the colliding molecules is able to surmount a centrifugal barrier the reaction is certain. This is known as the capture approximation; to emphasise this conceptual
difference between simulations and theoretical models, we use the word *capture* instead of *collision* to refer to theory-based results.

The interaction between two identical polar molecules is usually written as

$$V = \frac{(\boldsymbol{d}_1 \cdot \boldsymbol{d}_2) - 3(\boldsymbol{d}_1 \cdot \boldsymbol{n})(\boldsymbol{d}_2 \cdot \boldsymbol{n})}{r^3} + U(r), \qquad (5)$$

where $\boldsymbol{d}_1$ and $\boldsymbol{d}_2$ are the dipole moment vectors of the molecules, $\boldsymbol{n}$ is the unit vector along the distance vector $\boldsymbol{r}$ connecting the centres of mass of the molecules, and $U(r)$ is a spherically symmetric potential, usually proportional to
$r^{-6}$. The capture rate constant for such a potential can only

be calculated numerically (Maergoiz et al., 1996c). In the present section we consider only the isotropic part $U(r)$ of the interaction described by Eq. (5), the effect of anisotropic part (first term in Eq. (5)) is discussed in Appendix A. Then, the Langevin model of capture (Langevin, 1905) can be
used to calculate the critical impact parameter beyond which point-like colliding particles in vacuum will escape from each other. Here, the motion of the two colliding molecules is reduced to a one-body problem in an external central field by using an effective potential containing dispersion and cen-
trifugal terms (Landau and Lifshitz, 1976),

$$U_{\text{eff}}(r) = U(r) + \frac{L^2}{2\mu r^2}, \qquad (6)$$

where $r$ is the distance of the colliding body from the centre of the field, $L$ the angular momentum. Both the total energy of the system (which equals the initial translational energy
$\mu v^2/2$ at $r \to \infty$) and the angular momentum are conserved. The centrifugal term introduces an energy barrier, and for a successful capture at the barrier ($r = r_{\text{max}}$) the translational energy $\mu \dot{r}_{\text{max}}^2/2$ has to be positive. Since the angular momentum equals $L = \mu v b$, the condition for $b^2$ to ensure a cap-
ture is

$$b^2 < r_{\text{max}}^2 \left(1 - \frac{2U(r_{\text{max}})}{\mu v^2}\right). \qquad (7)$$

In case of a simple attractive potential (repulsive forces can be neglected, as the studied velocities are relatively low),

$$U(r) = -\epsilon \left(\frac{r_0}{r}\right)^6, \qquad (8)$$

the square of the critical impact parameter can be written as

$$b_{\text{c}}^2 = \left(\frac{27\epsilon}{2\mu v^2}\right)^{1/3} r_0^2. \qquad (9)$$

It is preferable to consider the squared value of $b$, since the capture cross section is calculated as $\sigma_{\text{c}} = \pi b_{\text{c}}^2$. It is important to note that in the Langevin model, the total energy is
divided strictly to the translational and potential energy, the internal degrees of freedom of the two bodies are considered to be completely decoupled, i.e. exchange of translational energy to rotations and vibrations that will occur in a real molecule is completely neglected.

## 2.5  Brownian model of aerosol coagulation

In the study by Kürten et al. (2014), a model of Brownian coagulation in a field of force (Sceats, 1986, 1989) was used to estimate the collision enhancement factor for neutral cluster formation involving sulfuric acid in a free
molecule regime (Chan and Mozurkewich, 2001). The model is based on solving the Fokker–Planck equation for a pair of Brownian particles whose motion is determined by a thermal random force (Sceats, 1986). In the paper by Chan and

Mozurkewich (2001), the Hamaker constant describing the strength of the van der Waals interaction was fitted to experiments with uncharged $H_2SO_4/H_2O$ particles with diameters of 49–127 nm, yielding a collisions enhancement factor value of 2.3 at 300 K. Although the Hamaker constant is usually considered to be size-independent, there may be enhanced interaction for very small particle sizes, with radii of the order of 1 nm (Pinchuk, 2012).

For the attractive potential described by Eq. (8), the collision enhancement factor over the kinetic gas theory rate, $W_B = \beta_B/\beta_{HS}$, from the Brownian coagulation model in the free molecule limit can be written as (Sceats, 1989)

$$W_B = \left(\frac{3\epsilon}{k_B T}\right)^{1/3}\left(\frac{r_0}{2R}\right)^2 \exp\left(\frac{1}{3}\right). \tag{10}$$

To compare the Brownian and the Langevin models to atomistic simulation results using the OPLS force field, we have determined the attractive potential parameters $\epsilon$ and $r_0$ to be equal to $-2\Delta F$ and the sulfur-sulfur distance at which the PMF reaches its minimum, respectively.

It should be noted that the model of Brownian coagulation does not describe the correct transport physics of collisions of molecules in the gas phase. For a discussion on the transition from the free molecular (ballistic) regime to the continuum (diffusive) regime, see e.g. Ouyang et al. (2012).

## 3   Results and Discussion

The statistics of the collision probabilities as a function of the impact parameter and relative velocity, $P(b,v)$, obtained from the atomistic simulations are shown as a heat map in Fig. 5 where white indicates a certain collision event (defined by the formation of one or more hydrogen bonds) and black indicates zero collision probability. The critical impact parameter (above which a collision unlikely occurs) has a negative exponential dependency on the relative velocity, which corresponds well with the Langevin model. A more detailed plot is provided in Fig. B1, where the sigmoidal probability curves are shown for each velocity separately. Predominantly, the collision probability approaches to unity with lower relative velocity and shorter impact parameter. However, the collision probability decreases slightly at high relative velocities where rapid re-dissociation of the complex can be caused by high kinetic energy and slow redistribution of the energy to vibrational modes of the formed cluster. At the slowest velocity of 50 ms$^{-1}$, and small values of the impact parameter, the collision probability is also reduced. This happens because in some cases the fluctuations in the intermolecular energy, just as they come within interaction range, are sufficient to exceed the very small initial translational energies of the colliding molecules, effectively repelling them (see Appendix C for more detailed discussion).

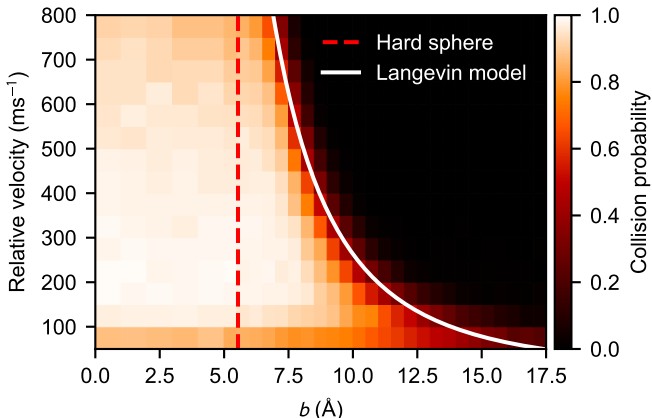

**Figure 5.** Heat map of the collision probability of sulfuric acid molecules plotted as a function of impact parameter $b$ and relative velocity $v$ obtained from molecular dynamics simulation. The impact parameter equivalent to the hard-sphere collision area ($b = 2R$) and the critical impact parameter obtained from the Langevin capture model (Eq. (9)) are indicated by the dashed red and the solid white lines, respectively.

The dynamical collision cross section, obtained from the integral over the collision probability functions,

$$\sigma_d(v) = \pi \int_0^\infty db^2 P(b,v), \tag{11}$$

is consistently decreasing with relative velocity $v$. Even though it can be seen in Fig. 5, and especially in Fig. B1, that values of $\sigma_d$ are smaller than the corresponding Langevin capture cross sections $\sigma_c$, the velocity dependence of the change in $\sigma_d$ is in very close agreement with the Langevin model solution

$$\frac{\sigma_c(v)}{\sigma_c(v_0)} = \left(\frac{v}{v_0}\right)^{-2/3}, \tag{12}$$

where $v_0$ is a reference velocity, as shown in Fig. 6. The importance of contributions from long-ranged interactions to the collision cross section is evident, as $\sigma_c$ is proportional to $v^{-4/n}$, for interactions decaying with $r^{-n}$. Furthermore, as can be seen in Fig. B1, the critical impact parameters from the Langevin model are matching rather well with the tails of the simulated collision probability curves, the intersection is located without exception at $P(b,v) \approx 0.2$.

The discrepancy between $\sigma_d$ and $\sigma_c$ is the result of the assumptions made in the Langevin model, where the capture is considered to be orientation-independent and the particles do not have any internal structure. If the anisotropy of the dipole–dipole potential is taken into account, as in Eq. (5), the capture cross section will be reduced (this has been estimated in Appendix A using a numerical approach provided by Maergoiz et al. (1996c)). However, if two molecules are

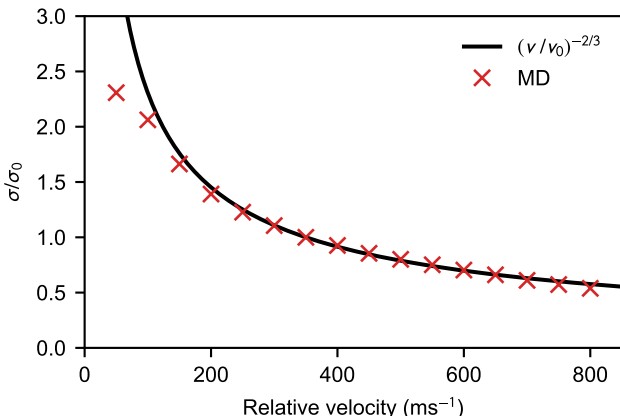

**Figure 6.** Ratio between collision cross section $\sigma$ and a reference value $\sigma_0$ as a function of relative velocity. Here the reference value $\sigma_0$ is the cross section corresponding to relative velocity $v_0 = 350$ ms$^{-1}$. The red crosses show the ratio of dynamical cross sections obtained from MD simulations, and the solid black line shows the relation predicted by the Langevin model in Eq. (12).

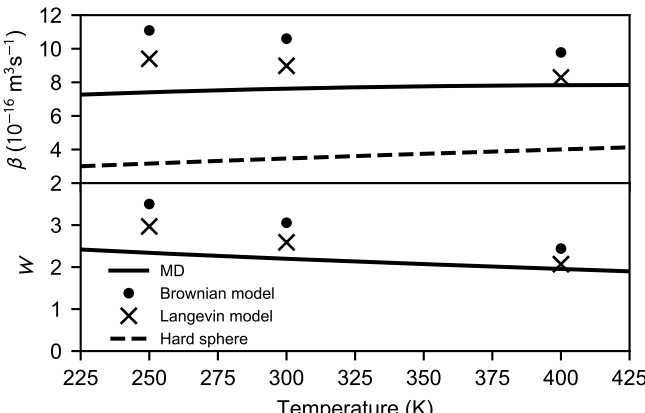

**Figure 7.** Collision rate coefficient $\beta$ (upper panel) and the enhancement factor $W$ (lower panel) as a function of temperature calculated for the hard-sphere, MD, Langevin and Brownian approaches.

able to move rather close to each other, translational energy can be transfered to rotational and vibrational modes, and therefore the motion over the centrifugal barrier is hindered, and the critical impact parameter effectively reduced. Additionally, steric hindrance caused by intermolecular orientations incompatible with the formation of hydrogen bonds will also lower the collision probability. Due to coupling, steric hindrance and other dynamical effects, the ratio between the cross sections $\sigma_d$ and $\sigma_c$ is on average 0.82 for the collision of two sulfuric acid molecules.

The canonical collision rate coefficient can be calculated similarly as Eq. (1), but since the collision probabilities $P(b,v)$ obtained from the atomistic simulations depend on both the velocity and the impact parameter, the MD based collision rate coefficient is calculated by integrating over both the relative velocity distribution $f(v)$ and $b^2$ as

$$\beta_{\mathrm{MD}} = \pi \int\limits_0^\infty \mathrm{d}v \int\limits_0^\infty \mathrm{d}b^2 \, v \, f(v) \, P(b,v). \qquad (13)$$

While the thermal velocity distribution $f(v)$ of the colliding molecules can be altered freely to correspond with an arbitrary temperature, the effect of the internal motion to the collision probability function is not necessarily temperature-invariant. However, in Appendix B it has been shown that a moderate change (simulations carried out at 250 and 400 K) in the internal kinetic energy does not effect the collision probabilities significantly. We therefore used the collision probability distributions calculated at 300 K to compute the collision rate coefficients for the atmospherically relevant temperature range $T = 225 - 425$ K (see Fig. 7).

In case of the Langevin model (Eq. (9)) the expression for the canonical capture rate coefficient can be simplified to

$$\beta_{\mathrm{L}} = \pi \int\limits_0^\infty \mathrm{d}v \, v \, f(v) \, b_{\mathrm{c}}^2(v). \qquad (14)$$

Since the potential of mean force is required for the Langevin model, the coefficients are calculated only at 250, 300 and 400 K (see Appendix B for further details).

In addition to the coefficients obtained by different approaches, we examine the enhancement factor $W$ relative to the kinetic gas theory rate expressed in Eq. (1), where a hard-sphere radius $R = 2.77$ Å was calculated from the bulk liquid density of sulfuric acid, $\rho = 1830$ kgm$^{-3}$, assuming a volume fraction of one. Thus, after performing the integration, the enhancement factor obtained using the Langevin model can be expressed analytically as

$$W_{\mathrm{L}} = \frac{\beta_{\mathrm{L}}}{\beta_{\mathrm{HS}}} = \Gamma\left(\frac{2}{3}\right)\left(\frac{2\epsilon}{k_{\mathrm{B}}T}\right)^{1/3}\left(\frac{r_0}{2R}\right)^2, \qquad (15)$$

where $\Gamma(x)$ denotes the Gamma function, and $\epsilon$ and $r_0$ are the parameters based on the potential of mean force between two sulfuric acid molecules in vacuum, using the functional form in Eq. (8). The enhancement factor from the Brownian model of aerosol coagulation, $W_{\mathrm{B}}$, was calculated analytically from Eq. (10).

We find that $W_{\mathrm{MD}} \approx 2.20$, $W_{\mathrm{L}} \approx 2.59$, and $W_{\mathrm{B}} \approx 3.06$ at 300 K. Both the collision rate coefficients and the enhancement factors are shown in Fig. 7. As the hard-sphere collision rate is linearly increasing as the molecular velocity is proportional to $\sqrt{T}$, the rate coefficient based on atomistic simulations is only slightly increasing with temperature due to the exponential narrowing of the collision cross section as the relative velocity increases. However, the dependency

is different in case of the Langevin model as the rate coefficient is decreasing with increasing temperature. This is due to the neglect of the anisotropy of the dipole-dipole interactions in the Langevin model: at higher temperatures, the effect of anisotropy becomes less important and therefore the model overestimates the collision rate less, compared to lower temperatures. The same effect can be observed in the Brownian coagulation model. Indeed, as seen from Fig. 7, the enhancement factors based on the theoretical models reach better agreement with the factor obtained from atomistic simulations as the temperature rises. As a result of the above-mentioned reasons, the enhancement factor over the hard-sphere collision rate coefficient is lower at higher temperatures regardless of the chosen approach.

The enhancement factor obtained by atomistic simulations is in very good agreement with the kinetic modelling on recent experimental results of formation of atmospheric sulfuric acid dimers (Kürten et al., 2014) and small clusters of sulfuric acid, dimethylamine and water (Lehtipalo et al., 2016; Kürten et al., 2018). In these studies, the enhancement factor was estimated to be 2.3–2.7 using the Brownian coagulation model and Van der Waals interactions fitted to experiment, as mentioned earlier, whereas according to our molecular model of long-range interaction the basic Brownian model using the attractive part of the potential of mean force overestimates the rate enhancement factor by about 40 %.

## 4   Conclusions

In summary, we have benchmarked two classical force fields against experimental and ab initio data and determined that the OPLS force field by Loukonen et al. was able to describe the geometry and vibrational spectra of the isolated sulfuric acid molecule, as well as the geometry and binding free energy of the sulfuric acid dimer. We studied the statistics of collisions of sulfuric acid molecules in vacuum by molecular dynamics simulations and compared our results against simple theoretical models. We have found that the effective collision cross section of two $H_2SO_4$ molecules, as described by the OPLS force field, is significantly larger than the hard-sphere diameter assigned to the molecule based on the liquid density of sulfuric acid. As a consequence, we find the collision coefficient for sulfuric acid molecules is enhanced by a factor 2.2, compared to kinetic gas theory at 300 K. This enhancement factor obtained from atomistic simulation is consistent with the discrepancy observed between experimental formation rates of clusters containing sulfuric acid and rates calculated using hard sphere kinetics. At a temperature range from 250 to 400 K, the rate enhancement factor is monotonously decreasing with increasing temperature, however the drop is less than 20 %. The velocity dependence of the simulated dynamical collision cross section is in good agreement with the Langevin model solution. We also note that the enhancement factor obtained from the Langevin model using the attractive part of the intermolecular potential is a bit overestimated due to the imperfect treatment of the dipole-dipole interaction, yet in the atmospherically relevant temperature range the factor is within 30 % of the result from the atomistic simulation, at a fraction of the computational cost.

In the future, the atomistic collision modelling approach presented in this work can be applied to other atmospherically relevant molecules, clusters, or ions, exhibiting dipoles of varying magnitude – and in some cases several times larger than the one of the sulfuric acid molecule – to help understand the effect of long-range interactions in cluster formation rates. However, before we can quantitatively assess the influence of collision rate enhancement on atmospherical new particle formation rates obtained from cluster dynamics models (for example ACDC (McGrath et al., 2012)), it is necessary to obtain the enhancement factors for all the relevant collisions between clusters of different sizes and composition, as the pathway for growth may change – a formidable task, even if only the simplest acid-base clusters were considered. Future work therefore should also be aimed at finding simple models for predicting approximate rate enhancements, based on just a few physico-chemical properties, such as molecular structures, dipole moments or charge distributions, of the interacting molecules and/or clusters.

## Appendix A:   Effect of anisotropy on the dipole–dipole capture rate

Maergoiz et al. (1996c), using classical trajectory integration, calculated the capture rate constant when two identical polar molecules interact through a potential containing anisotropic ($\propto r^{-3}$) and isotropic ($\propto r^{-6}$) terms,

$$V = \frac{(\boldsymbol{d}_1 \cdot \boldsymbol{d}_2) - 3(\boldsymbol{d}_1 \cdot \boldsymbol{n})(\boldsymbol{d}_2 \cdot \boldsymbol{n})}{r^3} - \frac{C}{r^6}, \qquad (A1)$$

where $\boldsymbol{d}_1$ and $\boldsymbol{d}_2$ are the dipole moments vectors of the molecules, $\boldsymbol{n}$ is the unit vector along the distance vector $\boldsymbol{r}$ connecting the centres of mass of the molecules and $C$ is the isotropic interaction constant.

As in Eq. (1), the capture rate coefficient in an anisotropic field is given by

$$\beta_{\mathrm{aniso}} = \sqrt{\frac{8 k_{\mathrm{B}} T}{\pi \mu}} \, \overline{\sigma}_{\mathrm{aniso}}, \qquad (A2)$$

where the thermal capture cross section can be calculated using a fitting function $\kappa(\theta, M)$ as

$$\overline{\sigma}_{\mathrm{aniso}} = \pi \left( \frac{d^2}{k_B T} \right)^{2/3} \theta^{1/6} \kappa(\theta, M), \qquad (A3)$$

where $d$ is the molecular dipole moment (for the OPLS model of sulfuric acid, $d = |\boldsymbol{d}_1| = |\boldsymbol{d}_2| = 3.07$  Debye).

Maergoiz et al. use two dimensionless parameters in their model:

$$\theta = \frac{Ck_BT}{d^4}, \tag{A4}$$

and

5 $$M = \frac{\mu d^{4/3}}{2I(k_BT)^{2/3}}. \tag{A5}$$

Based on our MD simulations using the OPLS force field, the average moment of inertia $I$ of a vibrating sulfuric acid molecule is $100.04$ amuÅ$^2$, which deviates slightly from values $100.66$ and $104.94$ amuÅ$^2$ obtained experimentally 10 (Kuczkowski et al., 1981) and from quantum chemical calculations (Zapadinsky et al., 2019), respectively. The fitting function is obtained from classical trajectory calculations and it is expressed as

$$\ln \kappa(\theta, M) = a_0 + \left( \frac{a_1^2 z}{\sinh(z)} + \frac{z^2}{36} \right)^{1/2}, \tag{A6}$$

15 where

$$z = a_2 + \ln \theta. \tag{A7}$$

The reported fitting parameters are (Maergoiz et al., 1996c)

$$a_0 = 0.2406 - 0.1596 \left( 1 + 1.9192 M^{0.9935} \right)^{-1}, \tag{A8a}$$

$$a_1 = 0.253 - 0.04573 \left( 1 + 1.1645 M^{0.6422} \right)^{-1}, \tag{A8b}$$

20 $$a_2 = 1.7617 + 0.9577 \left( 1 + 1.9192 M^{0.9935} \right)^{-1}. \tag{A8c}$$

Since all different long-range interactions are included in the attractive part of the potential of mean force between two sulfuric acid molecules (Eq. (8)), to exclude the dipole–dipole interaction from the isotropic interaction, the constant 25 $C$ is written as

$$C = (1 - f)\epsilon r_0^6, \tag{A9}$$

where $f$ is a factor denoting the relative magnitude of the anisotropic interaction between permanent dipoles with respect to the total interaction. Thus, the rate coefficient is con-30 trolled by the relative dipole–dipole interaction and the enhancement factor over the kinetic gas theory can be written as

$$W_{\mathrm{aniso}}(f) = \frac{\beta_{\mathrm{aniso}}(f)}{\beta_{\mathrm{HS}}} = \frac{\bar{\sigma}_{\mathrm{aniso}}(f)}{\pi(2R)^2}. \tag{A10}$$

Figure A1 shows the rate enhancement as a function of the 35 interaction factor $f$ at 300 K. As the anisotropic part does not contribute, i.e. $f = 0$, the enhancement factor is less than 4 % higher than the value obtained from the Langevin model (the statistical error of the thermal capture cross section is about 2 % (Maergoiz et al., 1996c)).

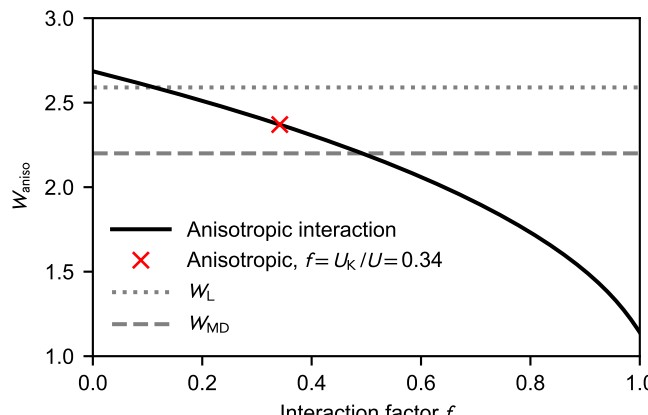

**Figure A1.** Enhancement factor $W_{\mathrm{aniso}}$ over the kinetic gas theory calculated from the anisotropic dipole–dipole collision cross section as a function of the interaction factor $f$ at 300 K (black solid line). The red cross denotes the case where the dipole–dipole contribution of the total interaction equals the Keesom equation Eq. (A11) ($W_{\mathrm{aniso}} = 2.41$). The enhancement factors obtained from the Langevin model and the MD simulations are shown as the dotted and dashed grey lines, respectively.

**Table A1.** The attractive potential parameters $\epsilon$ and $r_0$ for $H_2SO_4$–$H_2SO_4$ interaction based on the PMF calculations with the estimated anisotropic interaction factor $f = U_K/U$ and the corresponding enhancement factors calculated by the Langevin model $W_L$, anisotropic approach $W_{\mathrm{aniso}}(f)$ and atomistic simulations $W_{\mathrm{MD}}$.

| $T$ (K) | $\epsilon$ (eV) | $r_0$ (Å) | $f$ | $W_L$ | $W_{\mathrm{aniso}}(f)$ | $W_{\mathrm{MD}}$ |
|---|---|---|---|---|---|---|
| 250 | 0.69 | 4.1 | 0.33 | 2.97 | 2.73 | 2.34 |
| 300 | 0.55 | 4.1 | 0.34 | 2.59 | 2.37 | 2.20 |
| 400 | 0.37 | 4.1 | 0.38 | 2.07 | 1.87 | 1.95 |

Since we are unable to distinguish the actual dipole–dipole 40 interaction from the total attractive potential, we have estimated the interaction using the Keesom equation (see Fig. 3):

$$U_K = -\frac{d^4}{24\pi^2 \epsilon_0^2 k_B T r^6}. \tag{A11}$$

According to Eqs. (8) and (A11), about one third of the 45 attractive potential is due to dipole–dipole interaction, and consequently the enhancement factor is lower than for the isotropic field. The estimated rate enhancement factors at 250, 300 and 400 K are shown in Tab. A1. Thus, by taking into account the anisotropy of the intermolecular potential, 50 the estimated capture rate is in better agreement with the result obtained using atomistic simulations.

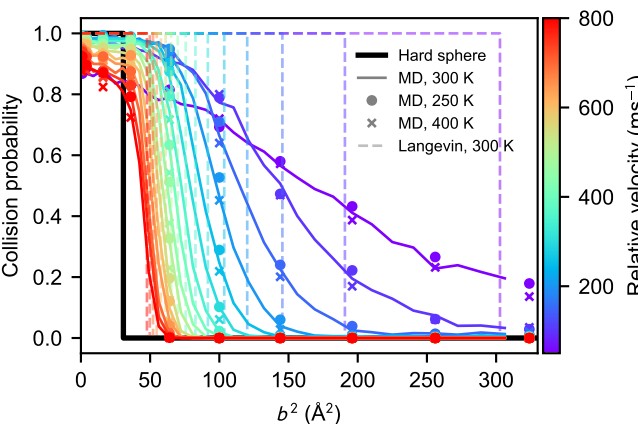

**Figure B1.** Collision probabilities of sulfuric acid molecules, as a function of the impact parameter squared, for different values of the relative velocity, obtained from molecular dynamics simulation at 300 K (solid coloured lines), at 250 K (coloured dots) and at 400 K (coloured crosses). The step-like collision probabilities for a hard-sphere model ($b^2 = (2R)^2$), or obtained from the Langevin capture model (Eq. (9)), are indicated by the solid black, and dashed coloured lines, respectively.

**Figure C1.** Time evolution of the intermolecular energy (red line) and the sulfur–sulfur distance (solid black line) in one MD trajectory with relative velocity $v = 50$ ms$^{-1}$ and $b = 0$ Å where a collision occurs (top) and one where the molecules are repelled at range (bottom). The initial translational kinetic energy of the molecules (1.27 meV) is indicated by the dashed black line.

## Appendix B: Temperature dependence of collision probabilities and interaction parameters

The canonical collision rate coefficient can be calculated from the collision probabilities obtained from atomistic simulation at arbitrary temperatures by shifting the Maxwell-Boltzmann relative velocity distribution, provided that changes in the internal motion of the molecules do not affect the collision probabilities. We have tested the effect of the different rotational and vibrational motion on the collision statistics in an atmospherically relevant temperature range by carrying out MD collision simulations for a subset of impact parameters $b$ where the molecules' atomistic velocities were drawn from Maxwell-Boltzmann distributions corresponding to 250 K and 400 K, instead of 300 K. As shown in Fig. B1, such moderate change in temperature indeed does not affect the collision probabilities between two sulfuric acid molecules.

In order to vary temperature in calculating the thermal collision rate coefficient using the Langevin approach, the potential of mean force between two sulfuric acid molecules was calculated at 250 K, 300 K, and 400 K, and the parameters describing the attractive intermolecular interaction (Eq. (8)) are reported in Table A1.

## Appendix C: Intermolecular repulsion at low velocities in MD simulations

As shown in Fig.s 5 and B1, for small values of the impact parameter and initial relative velocity between two colliding molecules in the atomistic simulations, the collision probability can be considerably smaller than unity, which seems counter-intuitive at first. This is due to the fact that the intermolecular interaction is anisotropic and the molecules are rotating, which can lead to instantaneous repulsion even at distances where the intermolecular potential of mean force is slightly attractive. If the initial translational kinetic energy is low enough, the temporary fluctuations in intermolecular energy can alter the translational motion and eventually lead to a definitive separation of the molecules.

This process is illustrated in Fig. C1, which shows the evolution of the intermolecular energy and distance in one trajectory with $b = 0$ Å and $v = 50$ ms$^{-1}$ where a collision occurs and a second one where the molecules are repelled at range. While in both cases the fluctuation of the intermolecular energy exceeds the initial translational energy of 1.27 meV, in the trajectory exhibiting a repulsion, the large positive energy fluctuations are longer lived and dominate the interaction.

*Author contributions.* RH and BR planned the simulation setup and performed benchmark calculations. BR carried out collision and metadynamics simulations and wrote the first draft of the paper. RH and EZ provided the theoretical framework. RH analyzed the simulation data. TK and HV helped plan the project and all authors contributed to writing the manuscript.

*Competing interests.* The authors declare that they have no conflict of interest.

*Acknowledgements.* This work was supported by the European Research Council (project 692891 DAMOCLES), Academy of Finland (Academy Research Fellow project 1266388 and ARKTIKO project 285067 ICINA), and University of Helsinki, Faculty of Science ATMATH project. Computational resources were provided by CSC–IT Center for Science, Ltd., Finland.

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
