# Peer review of "Roope Halonen1, Evgeni Zapadinsky1, Theo Kurtén2, Hanna Vehkamäki1, and Bernhard Reischl1"

_Atmospheric Chemistry and Physics, 2019_

## Referee Comment (RC1) · Anonymous Referee #1 · 23 Jun 2019

This manuscript discusses calculation of the collision rate between two sulfuric acid molecules in the gas phase using molecular dynamics calculations. The authors find that the binding rate/collision rate is $\sim$ a factor of 2.2 larger than would be expected based on hard sphere calculations. More detailed collision rate calculations are very important for molecules involved in new particle formation, as the resulting collision rate coefficients can be input into models of new particle formation and growth. This improves the accuracy and physical grounding of NPF models.

I think this study is quite promising, very well-written, and the manuscript is easy to follow. However, I do think that calculation of the enhancement factor at a single temperature is of limited use; atmospheric systems are not all at a single temperature, and it is equally important to determine if the collision rate coefficient increases or decreases with temperature (i.e. its derivative). Fortunately, this should be possible to address in revision, and there are similar recent works (in very different systems) the authors could follow to address this issue, as noted below.

1. Section 2.3. and Figure 5. The methods the authors use for binding rate coefficient calculations are nearly identical to those recently used by Yang, Goudeli, and Hogan (2018). Condensation and dissociation rates for gas phase metal clusters from molecular dynamics trajectory calculations. The Journal of Chemical Physics. 164304. It would be good to acknowledge that this approach has been utilized previously. In addition, in presenting results, Yang et al (2018) show collision probability contour plots as a function of (b,v). I find these more intuitive to follow than Figure 5, thus I would recommend the authors look into providing these results as a contour plot.

2. Section 2.5. The collision between two un-ionized molecules in the gas phase at atmospheric pressure conditions is absolutely a free molecular process, and there is really no reason to compare the enhancement factor to the collision rate enhancement factor that applies in the continuum (diffusive or Brownian) limit. I would recommend removing it or altering the discussion to note that this calculation is simply included for reference, as it is not grounded in the correct transport physics for gas phase, molecular scale collisions. How the enhancement factor changes from the free molecular (ballistic) to transition to continuum (diffusive) regimes is discussed in Ouyang, Gopalakrishnan, and Hogan. (2012) Nanoparticle collisions in the gas phase in the presence of singular contact potentials. The Journal of Chemical Physics. 064316.

3. Results and Discussion. I think a key issue to address in the manuscript is that presently the enhancement factor is only calculated at a single temperature. The evolution of it with temperature is of equal interest. Again, following Yang et al (2018) (Figures 5 and 6 of their work, in particular), I think this can be addressed to lead to an improved manuscript. First, using equation (10) of the current manuscript, the authors

can vary the "translational" Temperature by shifting the Maxwell-Boltzmann distribution to see how the enhancement factor changes. Presumably, the enhancement factor decreases with increasing translational temperature, but it is not clear whether the actual collision rate coefficient increases or decreases with increasing temperature (in the hard sphere model it does, but many gas phase reactions have decreasing rates with increasing temperature). Of course, this approach neglects the changes in internal energies of the molecules, and adjusting internal energies the more time consuming effort of rerunning simulations with different initial equilibration temperatures. Still, I would encourage the authors to do these calculations using at least one more temperature, to see how different they are from the results of simply shifting the equation (10) Maxwell-Boltzmann distribution. An enhancement factor calculated at a single temperature is of limited use if the temperature sensitivity is not explored and discussed.

4. Conclusions: After addressing comment 3 it is important to determine if the Langevin model is accurate at all temperatures, or just within 20% of calculations near 300 K. In addition, I think it would be good to discuss the implications of calculations for new particle formation and growth models more explicitly.

---

## Referee Comment (RC2) · Anonymous Referee #2 · 10 Jul 2019

General Comments:

The authors calculate the collision rate of two sulfuric acid (SA) molecules in gas phase using atomistic molecular dynamics (MD) instead of the traditional hard sphere kinetic gas theory that is based on the diameter of sulfuric acid derived from its bulk liquid density. They benchmark two force fields for SA against ab initio results and conclude that an OPLS all-atom force field is better suited for the MD simulations. They find that the traditional kinetic gas theory underestimates the collision coefficient of two SA molecules by a factor of 2.2 compared to the MD simulations at 300 K. This discrepancy is consistent with empirical scaling used to match experimental new particle formation

(NPF) rates and with those from theoretical ones employing hard sphere kinetics. They also explore other simpler models for calculating collision coefficients such as Brownian coagulation and Langevin dynamics. They find that both simpler models perform better than hard sphere kinetics and that their accuracy depends on the velocity of the colliding sulfuric acid molecules.

The work is promising in that it charts a new way to incorporate accurate collision rates into NPF rate calculations. The collision rate corrections for other species involved in sulfate aerosol formation are presumably larger than those for sulfuric acid, making this work particularly important. However, the authors need to address one critical point before the manuscript's acceptance for publication. They have previously employed their Atmospheric Cluster Dynamics Code (ACDC) to calculate NPF rates for various sulfate aerosol systems.[McGrath, M. J., Olenius, T., Ortega, I. K., Loukonen, V., Paasonen, P., KurteÌAn, T., Kulmala, M., and VehkamaÌLki, H.: Atmospheric Cluster Dynamics Code: a Flexible Method for Solution of the Birth–Death Equations, Atmos. Chem. Phys., 12, 2345–2355, https://doi.org/10.5194/acp-12-2345-2012, http://www.atmos-chem-phys.net/12/2345/2012/, 2012.] ACDC uses collision rates from hard sphere kinetics and evaporation rates from quantum mechanically derived Gibbs free energies to calculate the population of clusters and NPF rates. It would be appropriate for the authors to demonstrate how the cluster populations and NPF would change using collision rates from atomistic MD simulations. Such a comparison will also put the current work in the greater context of calculating NPFs which is the ultimate goal of studies like the current one.

Specific Comments:

1. Page 1, line 19: Define "impact parameter"

2. Page 2, line 8-10: The following statement is important enough to warrant a more detailed discussion. "In fact, it has recently been found that collision coefficients obtained in this way had to be scaled by a factor 2.3–2.7 to predict kinetically limited

nucleation rates in agreement with experiment, for a system containing sulfuric acid, dimethylamine and water (KuÌĹrten et al., 2014; Lehtipalo et al., 2016; KuÌĹrten et al., 2018).

3. Page 2, line 11: Define "capture rate constant" and how it differs from "collision rate constant". If collision and capture rates are the same for the purposes of this work, the authors should stick with one or the other for the sake of clarity.

4. Page 3, Section 2.1: What are the exact forms of the Ding and Loukonen/OPLS force fields? What terms are included? Which one is more flexible? Including this information will be instructive to the reader.

5. Page 4, Figure 1: The authors should label the different hydrogen bonds in the figure to facilitate cross-referencing with the d[O...H] lines in Table 1.

6. Page 4, Table 1: It is curious that the authors benchmark the two force fields against a 2012 paper while more recent and more rigorous computational results should be available. The authors should reference other high quality works on the sulfuric acid dimer and justify their choice to use the 2012 paper as a benchmark.

7. Page 4, Table 1: It is curious why the authors use eV units for their $\Delta\Delta E$ values while the most commonly used unit is kcal mol-1.

8. Page 8, line 16: "diameters of 49-127nm" seems incorrect. Perhaps the units are wrong.
* * *

---

## Author Comment (AC1) · 5 Sep 2019

We thank referee #1 for the favourable assessment of our manuscript.

ad 1.
We thank the referee for bringing this very relevant paper to our attention – we will acknowledge it both in the introduction and the methods section. We have also prepared a 2D density map of collision probability as a function of impact parameter and relative velocity in a similar style as in the reference mentioned above (see Fig. 1). This will replace the original plot showing individual collision probabilities as function of impact parameter at a certain relative velocity in the results section. The original plot, including

the new data at 250 and 400 K (Fig. 2), will be moved to the appendix.

ad 2.

We agree with the referee – we mainly included this section because the Brownian coagulation model was used in the papers emphasising the discrepancy between experimental data and kinetic modelling of new particle formation (Kürten et al. (2014) and Lehtipalo et al. (2016)), but it was not explained in detail there. We will add the following "disclaimer" to the section: "It should be noted that the model of Brownian coagulation does not describe the correct transport physics of collisions of molecules in the gas phase."

ad 3 and 4.

We thank the referee for pointing out this limitation in our simulations and analyses. As the referee pointed out, temperature has a double effect on systems of colliding molecules: first, the Maxwell-Boltzmann distributions of relative velocities are different, and second, the rotational and vibrational motion of the molecules are different. In the calculation of collision coefficients from both the MD simulations as well as simpler models such as the Langevin approach, the first point can be addressed by carrying out the integration over the appropriate velocity distribution. However, the second point requires carrying out MD simulations at different temperature. In order to check the effect of temperature on the collision probabilities, we have rerun MD simulations with a subset of impact parameters with initial rotational and vibrational energies corresponding to 250 K and 400 K, as typical atmospheric processes will happen in this temperature range. These additional simulations indicate however that the differences in collision statistics at 250 K, 300 K, and 400 K, at a given relative velocity, are very small (see Fig. 2). We therefore used the collision probability distributions calculated at $T = 300$ K, obtained for more values of b, to compute the collision rate coefficients and enhancement factors for the temperature range $T = 250 - 400$ K. In this range, the collision rate coefficient is found to increase slightly with increasing temperature. The increase is smaller than in kinetic theory, where $\beta \sim T^{1/2}$ (see upper panel in Fig. 3).

[Figure]

The Langevin model also has an explicit temperature dependence in the Maxwell-Boltzmann distribution, as well as a temperature dependence of the intermolecular interaction parameter $\varepsilon$. To address this we have carried out additional PMF calculations for the $H_2SO_4$ pair at $T = 250$ and 400 K. The collision rate coefficient obtained from the Langevin model in this temperature range is found to decrease with increasing temperature (see upper panel in Fig. 3). This is due to the neglect of the anisotropy of the dipole-dipole interactions in the Langevin model: at higher temperatures, the effect of anisotropy becomes less important and therefore the model overestimates the collision rate less, compared to lower temperatures.

In the temperature range 250-400 K, the collision enhancement compared to kinetic theory decreases with temperature both for the MD simulations, and the Langevin model. At higher temperatures, the enhancement factor obtained from the Langevin model approaches the MD value, for reasons discussed above (see lower panel in Fig. 3). As we are interested in atmospheric new particle formation, we are not interested in temperatures outside of this range in the present work. We will add this discussion (including Fig. 3) to the manuscript and simulation details and Fig. 2 will be added to a new section in the appendix.

ad 4.
Regarding the very last comment on the effect of collision rate enhancement on new particle formation rates, we note that in cluster dynamics codes such as ACDC (McGrath et al., 2012) detailed balance is assumed, and therefore global changes to the collision rates obtained by application of an enhancement factor are compensated by the corresponding changes in evaporation rates. However, in complex systems, individually changing collision rates for reactions that are close to the kinetic limit can change the preferred pathway for cluster growth, leading to different cluster distributions and particle formation rates.

We will add the following paragraph to the conclusions. "However, before we can quantitatively assess the influence of collision rate enhancement on atmospherical new

particle formation rates obtained from cluster dynamics models (for example ACDC, McGrath et al., 2012), it is necessary to obtain the enhancement factors for all the relevant collisions between clusters of different sizes and composition, as the pathway for growth may change–a formidable task, even if only the simplest acid-base clusters were considered. Future work therefore should also be aimed at finding simple models for predicting approximate rate enhancements, based on just a few physico-chemical properties, such as molecular structures, dipole moments or charge distributions, of the interacting molecules and/or clusters."

Full figure captions:

Figure 1: Heat map of the collision probability of sulfuric acid molecules plotted as a function of impact parameter $b$ and relative velocity $v$ obtained from molecular dynamics simulation. The squared impact parameter equivalent to the hard-sphere collision area $(b^2 = (2R)^2)$ and the squared critical impact parameter obtained from the Langevin capture model (Eq. (9)) are indicated by the dashed red and the white lines, respectively.

Figure 2: Collision probabilities of sulfuric acid molecules, as a function of the impact parameter squared, for different values of the relative velocity, obtained from molecular dynamics simulation at 300 K (solid coloured lines), at 250 K (coloured dots) and at 400 K (coloured dots). The step-like collision probabilities for a hard-sphere model $(b^2 = (2R)^2)$, or obtained from the Langevin capture model (Eq. (9)), are indicated by the solid black, and dashed coloured lines, respectively.

[Figure]

**Fig. 1.** Heat map of the collision probability of sulfuric acid molecules plotted as a function of impact parameter b and relative velocity v obtained from molecular dynamics simulation. The squared impact

[Figure]

**Fig. 2.** Collision probabilities of sulfuric acid molecules, as a function of the impact parameter squared, for different values of the relative velocity, obtained from molecular dynamics simulation at 300 K (

[Figure]

**Fig. 3.** Collision rate coefficient $\beta$ (upper panel) and the enhancement factor W (lower panel) as a function of temperature calculated for the hard-sphere, MD, Langevin and Brownian approaches.

---

## Author Comment (AC2) · 5 Sep 2019

We thank referee #2 for the favourable review of our manuscript.

Regarding the general comment on the effect of collision rate enhancement on new particle formation rates, we note that in cluster dynamics codes such as ACDC (McGrath et al., 2012) detailed balance is assumed, and therefore global changes to the collision rates obtained by application of an enhancement factor are compensated by the corresponding changes in evaporation rates. However, in complex systems, individually changing collision rates for reactions that are close to the kinetic limit can change the preferred pathway for cluster growth, leading to different cluster distributions and

particle formation rates.

We will add the following paragraph to the conclusions. "However, before we can quantitatively assess the influence of collision rate enhancement on atmospherical new particle formation rates obtained from cluster dynamics models (for example ACDC, McGrath et al., 2012), it is necessary to obtain the enhancement factors for all the relevant collisions between clusters of different sizes and composition, as the pathway for growth may change–a formidable task, even if only the simplest acid-base clusters were considered. Future work therefore should also be aimed at finding simple models for predicting approximate rate enhancements, based on just a few physico-chemical properties, such as molecular structures, dipole moments or charge distributions, of the interacting molecules and/or clusters."

ad 1.
We will add the following clarification in the introduction: "[...] the impact parameter i.e. the perpendicular distance between the spheres' trajectories [...]"

ad 2.
We propose the following change to the sentence in the manuscript: "In fact, systematic discrepancies have been found between experimental particle formation rates and values predicted from kinetic modelling and cluster dynamics simulations, where hard-sphere collisions are assumed. Kürten et al. (2014) measured the kinetic formation rate of sulphuric acid dimers and found that an enhancement factor of 2.3 needed to be applied to the formation rate obtained from a kinetic model. Lehtipalo et al. (2016) and Kürten et al. (2018) have studied particle formation rates in systems containing sulphuric acid, dimethylamine and water and concluded that an enhancement factor of 2.7 and 2.3, respectively, was needed to match experimental particle formation rates."

ad 3.
We apologize for the lack of clarity regarding the difference between collision and capture. We will add the following paragraph to section 2.4: "As the collision rate in the

context of atomistic simulations is defined as the reaction rate of hydrogen bonding, the related theoretical models are often based on the assumption that if the trajectory of the colliding molecules is able to surmount a centrifugal barrier the reaction is certain. This is known as the capture approximation; to emphasise this conceptual difference between simulations and theoretical models, we use the word *capture* instead of *collision* to refer to theory-based results."

ad 4.

We have added the following description of the functional forms of the inter- and intramolecular potentials to the methods section: "In both force fields intermolecular interactions are described by the sum of Lennard-Jones potentials between atoms $i$ and $j$ with distance and energy parameters $\sigma_{ij}$ and $\varepsilon_{ij}$, and Coulomb interactions between the partial charges $q_i$ and $q_j$,

$$
\begin{aligned}
U_{\text{inter}} &= \sum_{i=1}^{N_1}\sum_{j=1}^{N_2} 4\varepsilon_{ij}\left[\left(\frac{\sigma_{ij}}{r_{ij}}\right)^{12} - \left(\frac{\sigma_{ij}}{r_{ij}}\right)^{6}\right] \\
&+ \sum_{i=1}^{N_1}\sum_{j=1}^{N_2} \frac{1}{4\pi\epsilon_0}\frac{q_i q_j}{r_{ij}}.
\end{aligned}
\tag{1}
$$

However, in the force field by Ding et al., the geometry of the individual molecule is simply constrained by harmonic potentials with force constants $k_{ij}$ between all pairs of atoms,

$$
U_{\text{intra}}^{\text{Ding}} = \sum_{i=1}^{N_1-1}\sum_{j=i+1}^{N_1} \frac{k_{ij}}{2}\left(r_{ij} - r_{ij}^0\right)^2,
\tag{2}
$$

while in OPLS the intramolecular interactions consist of the usual sum of two, three, and four-body potentials, i.e. harmonic bonds between covalently bonded atoms, harmonic angles between atoms separated by two covalent bonds, and torsions (dihedral angles)

[Figure]

between atoms separated by three covalent bonds,

$$
\begin{aligned}
U_{\text{intra}}^{\text{OPLS}} \quad = \quad & \sum_{i=1}^{N_{\text{bonds}}} \frac{k_i^b}{2} \left(r_i - r_i^0\right)^2 + \sum_{j=1}^{N_{\text{angles}}} \frac{k_j^\theta}{2} \left(\theta_j - \theta_j^0\right)^2 \\
+ \quad & \sum_{k=1}^{N_{\text{dihedrals}}} \sum_{n=1}^{4} \frac{V_n}{2} \left[1 + \cos(n\phi^k - \phi_n^k)\right] ."
\end{aligned}
\tag{3}
$$

ad 5.
We have numbered the hydrogen bonds for dimer structures a-d in Fig. 1 and added the corresponding hydrogen bond number to the hydrogen bond distance values in the table.

ad 6.
The paper by Temelso et al. (2012) is to our knowledge the only reference that contains detailed information on potential/electronic energies and hydrogen bond geometries for different conformers of the $H_2SO_4$ dimer, as well as the binding free energy. For the binding free energies, we found reasonable agreement between the study by Temelso et al. (2012) and more recent work by Elm et al. (2016) and Myllys et al. (2017), which we do mention in the manuscript. Note that Temelso et al. have obtained the binding free energy from Boltzmann-averaging over the four minimum energy dimer structures, while in the newer references only the global minimum energy structure has been considered.

ad 7.
In computational physics and chemistry, commonly used units of energy are eV, kJ/mol, kcal/mol, or $k_B T$. The unit used in the LAMMPS simulation in/output was eV, which is why this was the most natural choice for the manuscript. Since none of the important quantities we report, such as the collision rate coefficients, or enhancement factors, have the unit of energy, we think this should not be a major concern. However, to

accomodate all audiences, we will add the conversion factors to the manuscript: 1 eV $\approx 96.49$ kJ/mol $\approx 23.06$ kcal/mol $\approx 38.68 k_{\mathrm{B}} T$ at $T = 300$ K.

ad 8.

The values cited in our manuscript indeed correspond to the values given in the paper by Chan and Mozurkewich (2001), both in the abstract and in Fig. 5.

Full figure captions:

Figure 1: Four minimum energy structures for the sulfuric acid dimer (a–d) used to benchmark the force fields by Ding et al. (2003) and Loukonen et al. (2010) against ab initio calculations by Temelso et al. (2012) Sulfur atoms are yellow, oxygens red and hydrogens white. Hydrogen bonds are indicated by dotted lines and enumerated according to Tab. 1.
* * *
(a)

(b)

(c)

(d)

**Fig. 1.** Four minimum energy structures for the sulfuric acid dimer (a–d) used to benchmark the force fields by Ding et al. (2003) and Loukonen et al. (2010) against ab initio calculations by Temelso et al. (2

[Figure]

---

## Author Response (AR1)

**Response to referees' comments on manuscript "Rate enhancement in collisions of sulfuric acid molecules due to long-range intermolecular forces"**

Dear Editor,

We thank the two anonymous referees for the favorable review of our manuscript and the very good comments, based on which we have prepared a revised manuscript. We first briefly describe general changes to manuscript, before going through the referees' comments individually and show how they have been addressed in the revision. Our answers to referee comments are set in boldface. The complete revised manuscript, highlighting the differences to the original version, is included at the end of this response letter. We hope the changes are deemed satisfactory and the revised manuscript will be accepted for publication.

Kind Regards,

Roope Halonen, Evgeni Zapadinsky, Theo Kurtén, Hanna Vehkamäki and Bernhard Reischl

**1  General changes to the manuscript**

- Based on the comments by referee 1, we have studied the temperature dependence of the collision rate coefficients and enhancement factors for the molecular dynamics simulation, the Langevin model, and the Brownian coagulation model. This is reflected in additions and changes in the results section, as well as:

    - new Appendix B: Temperature dependence of collision probabilities and interaction parameters
    - new Figure 7: Collision rate coefficient $\beta$ (upper panel) and the enhancement factor $W$ (lower panel) as a function of temperature calculated for the hard-sphere, MD, Langevin and Brownian approaches.
    - new Figure B1: Collision probabilities of sulfuric acid molecules, as a function of the impact parameter squared, for different values of the relative velocity, obtained from molecular dynamics simulation at 300 K (solid coloured lines), at 250 K (coloured dots) and at 400 K (coloured crosses). The step-like collision probabilities for a hard-sphere model ($b^2 = (2R)^2$), or obtained from the Langevin capture model, are indicated by the solid black, and dashed coloured lines, respectively.
    - new Table A1: The attractive potential parameters $\epsilon$ and $r_0$ for $H_2SO_4$–$H_2SO_4$ interaction based on the PMF calculations with the estimated anisotropic interaction factor $f = U_K/U$ and the corresponding enhancement factors calculated by the Langevin model $W_L$, anisotropic approach $W_{aniso}(f)$ and atomistic simulations $W_{MD}$.

- We have recalculated all numerical values reported in the manuscript. A few values have changed slightly as a consequence, but there are no quantitative changes to our results.

- The analytical expression for the collision rate enhancement factor over kinetic theory, using the Langevin model, has been simplified (Eq. 15).

**2  Comments by Referee 1**

This manuscript discusses calculation of the collision rate between two sulfuric acid molecules in the gas phase using molecular dynamics calculations. The authors find that the binding rate/collision rate is $\sim$ a factor of 2.2 larger than would be expected based on hard sphere calculations. More detailed collision rate calculations are very important for molecules involved in new particle formation, as the resulting collision rate coefficients can be input into models of new particle formation and growth. This improves the accuracy and physical grounding of NPF models. I think this study is quite promising, very well-written, and the manuscript is easy to follow. However, I do think that calculation of the enhancement factor at a single temperature is of limited use; atmospheric systems are not all at a single temperature, and it is equally important to determine if the collision rate coefficient increases or decreases with temperature (i.e. its derivative). Fortunately, this should be possible to address in revision, and there are similar recent works (in very different systems) the authors could follow to address this issue, as noted below.

We thank referee 1 for the favorable assessment of our manuscript. The temperature dependence study has been carried out and is included in the revised manuscript.

Answers to Specific Comments:

1. Section 2.3. and Figure 5. The methods the authors use for binding rate coefficient calculations are nearly identical to those recently used by Yang, Goudeli, and Hogan (2018). Condensation and dissociation rates for gas phase metal clusters from molecular dynamics trajectory calculations. The Journal of Chemical Physics. 164304. It would be good to acknowledge that this approach has been utilized previously. In addition, in presenting results, Yang et al (2018) show collision probability contour plots as a function of (b,v). I find these more intuitive to follow than Figure 5, thus I would recommend the authors look into providing these results as a contour plot.

   **We thank the referee for bringing this very relevant paper to our attention–we have now acknowledged it both in the introduction ("Recently, Yang et al. (2018) have studied the condensation rate coefficients for Au and Mg clusters at various gas temperatures using molecular dynamics calculations.") and in the methods section ("The simulation setup is very similar to the one recently used by Yang et al. (2018).").**

   **We have also prepared a 2D density map of collision probability as a function of impact parameter and relative velocity in a similar style as in the reference mentioned above (see new Fig. 5). This replaces the original plot showing individual collision probabilities as function of impact parameter at a certain relative velocity in the results section. The original plot, including the new data obtained at 250 and 400 K has been moved to the new Appendix B, Fig. B1.:**
   **"The statistics of the collision probabilities as a function of the impact parameter and relative velocity, $P(b,v)$, obtained from the atomistic simulations are shown as a heat map in Fig. 5 where white indicates a certain collision event (defined by the formation of one or more hydrogen bonds) and black indicates zero collision probability. [...] A more detailed plot is provided in Fig. B1, where the sigmoidal probability curves are shown for each velocity separately."**

2. Section 2.5. The collision between two un-ionized molecules in the gas phase at atmospheric pressure conditions is absolutely a free molecular process, and there is really no reason to compare the enhancement factor to the collision rate enhancement factor that applies in the continuum (diffusive or Brownian) limit. I would recommend removing it or altering the discussion to note that this calculation is simply included for reference, as it is not grounded in the correct transport physics for gas phase, molecular scale collisions. How the enhancement factor changes from the free molecular (ballistic) to transition to continuum (diffusive) regimes is discussed in Ouyang, Gopalakrishnan, and Hogan. (2012) Nanoparticle collisions in the gas phase in the presence of singular contact potentials. The Journal of Chemical Physics. 064316.

   **We agree with the referee – we mainly included this section because the Brownian coagulation model was used in the papers emphasising the discrepancy between experimental data and kinetic modelling of new particle formation (Kürten et al. (2014) and Lehtipalo et al. (2016)), but it was not explained in detail there. To clarify this, we have added the following "disclaimer" to the end of the section: "It should be noted that the model of Brownian coagulation does not describe the correct transport physics of collisions of molecules in the gas phase. For a discussion on the transition from the free molecular (ballistic) regime to the continuum (diffusive) regime, see e.g. Ouyang et al. (2012)."**

3. Results and Discussion. I think a key issue to address in the manuscript is that presently the enhancement factor is only calculated at a single temperature. The evolution of it with temperature is of equal interest. Again, following Yang et al (2018) (Figures 5 and 6 of their work, in particular), I think this can be addressed to lead to an improved manuscript. First, using equation (10) of the current manuscript, the authors can vary the "translational" Temperature by shifting the Maxwell-Boltzmann distribution to see how the enhancement factor changes. Presumably, the enhancement factor decreases with increasing translational temperature, but it is not clear whether the actual collision rate coefficient increases or decreases with increasing temperature (in the hard sphere model it does, but many gas phase reactions have decreasing rates with increasing temperature). Of course, this approach neglects the changes in internal energies of the molecules, and adjusting internal energies the more time consuming effort of rerunning simulations with different initial equilibration

temperatures. Still, I would encourage the authors to do these calculations using at least one more temperature, to see how different they are from the results of simply shifting the equation (10) Maxwell-Boltzmann distribution. An enhancement factor calculated at a single temperature is of limited use if the temperature sensitivity is not explored and discussed.

**We thank the referee for pointing out this limitation in our simulations and analyses – we have carried out a study of the temperature dependence of the collision rate coefficients as well as the enhancement factors over kinetic theory in a range of atmospherically relevant temperatures. Including the new results has lead to many small additions and changes to the original manuscript. In the following, we summarize our additional study and the most important changes to the paper. For the full list of changes, we kindly refer to the manuscript with highlighted changes at the end of this response letter.**

**As the referee pointed out, temperature has a double effect on systems of colliding molecules: first, the Maxwell-Boltzmann distributions of relative velocities are different, and second, the rotational and vibrational motion of the molecules are different. In the calculation of collision coefficients from both the MD simulations as well as simpler models such as the Langevin approach, the first point can be addressed by carrying out the integration over the appropriate velocity distribution. However, the second point requires carrying out MD simulations at different temperature. In order to check the effect of temperature on the collision probabilities, we have rerun MD simulations with a subset of impact parameters with initial rotational and vibrational energies corresponding to 250 K and 400 K, as typical atmospheric processes will happen in this temperature range. These additional simulations indicate however that the differences in collision statistics at 250 K, 300 K, and 400 K, at a given relative velocity, are very small (see new Fig. B1). We therefore used the collision probability distributions calculated at $T = 300$ K, obtained for more values of $b$, to compute the collision rate coefficients and enhancement factors for the temperature range $T = 250$–400 K. In this range, the collision rate coefficient is found to increase slightly with increasing temperature. The increase is smaller than in kinetic theory, where $\beta \sim T^{1/2}$ (see upper panel in new Fig. 7).**

**The Langevin model also has an explicit temperature dependence in the Maxwell-Boltzmann distribution, as well as a temperature dependence of the intermolecular interaction parameter $\epsilon$. To address this we have carried out additional PMF calculations for the $H_2SO_4$ pair at $T = 250$ and 400 K. The collision rate coefficient obtained from the Langevin model in this temperature range is found to decrease with increasing temperature (see upper panel in Fig. 3). This is due to the neglect of the anisotropy of the dipole-dipole interactions in the Langevin model: at higher temperatures, the effect of anisotropy becomes less important and therefore the model overestimates the collision rate less, compared to lower temperatures.**

**In the temperature range 250-400 K, the collision enhancement compared to kinetic theory decreases with temperature both for the MD simulations, and the Langevin model. At higher temperatures, the enhancement factor obtained from the Langevin model approaches the MD value, for reasons discussed above (see lower panel in new Fig. 7). As we are interested in atmospheric new particle formation, we are not interested in temperatures outside of this range in the present work. We have added this discussion to the manuscript in the results and discussion, and simulation details have been added to a new section B in the appendix.**

**We have added the following paragraph in the results section:**
**"While the thermal velocity distribution $f(v)$ of the colliding molecules can be altered freely to correspond with an arbitrary temperature, the effect of the internal motion to the collision probability function is not necessarily temperature-invariant. However, in Appendix B it has been shown that a moderate change (simulations carried out at 250 and 400 K) in the internal kinetic energy does not affect the collision probabilities significantly. We therefore used the collision probability distributions calculated at 300 K to compute the collision rate coefficients for the atmospherically relevant temperature range $T$=225-425 K (see Fig. 7)."**

**We have added the following paragraph in the conclusions:**
**"In the temperature range from 250 to 400 K, the rate enhancement factor is monotonously decreasing with increasing temperature, however the drop is less than 20 %. The velocity dependence of the simulated dynamical collision cross section is in good agreement with the Langevin model solution. We also note that the enhancement factor obtained from the Langevin**

model using the attractive part of the intermolecular potential is a bit overestimated due to the imperfect treatment of the dipole-dipole interaction, yet in the atmospherically relevant temperature range the factor is within 30 % of the result from the atomistic simulation, at a fraction of the computational cost."

4. Conclusions: After addressing comment 3 it is important to determine if the Langevin model is accurate at all temperatures, or just within 20% of calculations near 300 K. In addition, I think it would be good to discuss the implications of calculations for new particle formation and growth models more explicitly.

   **Regarding the accuracy of the Langevin model, we have added the following statement to the conclusions:**
   **"[...] in the atmospherically relevant temperature range the [Langevin model enhancement] factor is within 30 % of the result from the atomistic simulation, at a fraction of the computational cost."**

   **Regarding the very last comment on the effect of collision rate enhancement on new particle formation rates, we note that in cluster dynamics codes such as ACDC (McGrath et al., 2012) detailed balance is assumed, and therefore global changes to the collision rates obtained by application of an enhancement factor are compensated by the corresponding changes in evaporation rates. However, in complex systems, individually changing collision rates for reactions that are close to the kinetic limit can change the preferred pathway for cluster growth, leading to different cluster distributions and particle formation rates.**

   **We have added the following paragraph to the conclusions:**
   **"However, before we can quantitatively assess the influence of collision rate enhancement on atmospherical new particle formation rates obtained from cluster dynamics models (for example ACDC (McGrath et al., 2012)), it is necessary to obtain the enhancement factors for all the relevant collisions between clusters of different sizes and composition, as the pathway for growth may change – a formidable task, even if only the simplest acid-base clusters were considered. Future work therefore should also be aimed at finding simple models for predicting approximate rate enhancements, based on just a few physico-chemical properties, such as molecular structures, dipole moments or charge distributions, of the interacting molecules and/or clusters."**

**3   Comments by Referee 2**

The authors calculate the collision rate of two sulfuric acid (SA) molecules in gas phase using atomistic molecular dynamics (MD) instead of the traditional hard sphere kinetic gas theory that is based on the diameter of sulfuric acid derived from its bulk liquid density. They benchmark two force fields for SA against ab initio results and conclude that an OPLS all-atom force field is better suited for the MD simulations. They find that the traditional kinetic gas theory underestimates the collision coefficient of two SA molecules by a factor of 2.2 compared to the MD simulations at 300 K. This discrepancy is consistent with empirical scaling used to match experimental new particle formation (NPF) rates and with those from theoretical ones employing hard sphere kinetics. They also explore other simpler models for calculating collision coefficients such as Brownian coagulation and Langevin dynamics. They find that both simpler models perform better than hard sphere kinetics and that their accuracy depends on the velocity of the colliding sulfuric acid molecules.

   The work is promising in that it charts a new way to incorporate accurate collision rates into NPF rate calculations. The collision rate corrections for other species involved in sulfate aerosol formation are presumably larger than those for sulfuric acid, making this work particularly important. However, the authors need to address one critical point before the manuscript's acceptance for publication. They have previously employed their Atmospheric Cluster Dynamics Code (ACDC) to calculate NPF rates for various sulfate aerosol systems.[McGrath, M. J., Olenius, T., Ortega, I. K., Loukonen, V., Paasonen, P., Kurtén, T., Kulmala, M., and Vehkamäi, H.: Atmospheric Cluster Dynamics Code: a Flexible Method for Solution of the Birth–Death Equations, Atmos. Chem. Phys., 12, 2345–2355, https://doi.org/10.5194/acp-12-2345-2012, http://www.atmos-chem-phys.net/12/2345/2012/, 2012.] ACDC uses collision rates from hard sphere kinetics and evaporation rates from quantum mechanically derived Gibbs free energies to calculate the population of clusters and NPF rates. It would be appropriate for the authors to demonstrate how the cluster populations and NPF would change using collision rates from atomistic MD simulations. Such a comparison will also put the current work in the greater context of calculating NPFs which is the ultimate goal of studies like the current one.

We thank referee **2** for the favourable review of our manuscript. Regarding the general comment on the effect of collision rate enhancement on new particle formation rates, we note that in cluster dynamics codes such as ACDC (McGrath et al., 2012) detailed balance is assumed, and therefore global changes to the collision rates obtained by application of an enhancement factor are compensated by the corresponding changes in evaporation rates. However, in complex systems, individually changing collision rates for reactions that are close to the kinetic limit can change the preferred pathway for cluster growth, leading to different cluster distributions and particle formation rates.

We have added the following paragraph to the conclusions:
"However, before we can quantitatively assess the influence of collision rate enhancement on atmospherical new particle formation rates obtained from cluster dynamics models (for example ACDC, McGrath et al., 2012), it is necessary to obtain the enhancement factors for all the relevant collisions between clusters of different sizes and composition, as the pathway for growth may change–a formidable task, even if only the simplest acid-base clusters were considered. Future work therefore should also be aimed at finding simple models for predicting approximate rate enhancements, based on just a few physico-chemical properties, such as molecular structures, dipole moments or charge distributions, of the interacting molecules and/or clusters."

Answers to Specific Comments:

1. Page 1, line 19: Define "impact parameter"

   **We have added the following clarification in the introduction: "[...] the impact parameter i.e. the perpendicular distance between the spheres' trajectories [...]"**

2. Page 2, line 8-10: The following statement is important enough to warrant a more detailed discussion. "In fact, it has recently been found that collision coefficients obtained in this way had to be scaled by a factor 2.3–2.7 to predict kinetically limited nucleation rates in agreement with experiment, for a system containing sulfuric acid, dimethylamine and water (Kürten et al., 2014; Lehtipalo et al., 2016; Kürten et al., 2018)."

   **We have made the following change to the sentence in the manuscript: "In fact, systematic discrepancies have been found between experimental particle formation rates and values predicted from kinetic modelling and cluster dynamics simulations, where hard-sphere collisions are assumed. Kürten et al. (2014) measured the kinetic formation rate of sulphuric acid dimers and found that an enhancement factor of 2.3 needed to be applied to the formation rate obtained from a kinetic model. Lehtipalo et al. (2016) and Kürten et al. (2018) have studied particle formation rates in systems containing sulphuric acid, dimethylamine and water and concluded that an enhancement factor of 2.7 and 2.3, respectively, was needed to match experimental particle formation rates."**

3. Page 2, line 11: Define "capture rate constant" and how it differs from "collision rate constant". If collision and capture rates are the same for the purposes of this work, the authors should stick with one or the other for the sake of clarity.

   **We apologize for the lack of clarity regarding the difference between collision and capture. We have added the following paragraph to section 2.4: "As the collision rate in the context of atomistic simulations is defined as the reaction rate of hydrogen bonding, the related theoretical models are often based on the assumption that if the trajectory of the colliding molecules is able to surmount a centrifugal barrier the reaction is certain. This is known as the capture approximation; to emphasise this conceptual difference between simulations and theoretical models, we use the word *capture* instead of *collision* to refer to theory-based results."**

4. Page 3, Section 2.1: What are the exact forms of the Ding and Loukonen/OPLS force fields? What terms are included? Which one is more flexible? Including this information will be instructive to the reader.

   **We have added the following description of the functional forms of the inter- and intramolecular potentials to the methods section:**
   **"In both force fields intermolecular interactions are described by the sum of Lennard-Jones potentials between atoms $i$ and $j$ separated by a distance $r_{ij}$, with distance and energy parameters**

$\sigma_{ij}$ and $\varepsilon_{ij}$, and Coulomb interactions between the partial charges $q_i$ and $q_j$,

$$
\begin{aligned}
U_{\text{inter}} \quad = \quad & \sum_{i=1}^{N_1}\sum_{j=1}^{N_2} 4\varepsilon_{ij} \left[ \left(\frac{\sigma_{ij}}{r_{ij}}\right)^{12} - \left(\frac{\sigma_{ij}}{r_{ij}}\right)^{6} \right] \\
+ \quad & \sum_{i=1}^{N_1}\sum_{j=1}^{N_2} \frac{1}{4\pi\epsilon_0} \frac{q_i q_j}{r_{ij}}.
\end{aligned}
\tag{1}
$$

However, in the force field by Ding et al., the geometry of the individual molecule is simply constrained by harmonic potentials with force constants $k_{ij}$ between all pairs of atoms,

$$
U_{\text{intra}}^{\text{Ding}} = \sum_{i=1}^{N_1-1}\sum_{j=i+1}^{N_1} \frac{k_{ij}}{2} \left(r_{ij} - r_{ij}^0\right)^2,
\tag{2}
$$

while in OPLS the intramolecular interactions consist of the usual sum of two, three, and four-body potentials, i.e. harmonic bonds between covalently bonded atoms, harmonic angles $\theta$ between atoms separated by two covalent bonds, and torsions (dihedral angles $\phi$) between atoms separated by three covalent bonds,

$$
\begin{aligned}
U_{\text{intra}}^{\text{OPLS}} \quad = \quad & \sum_{i=1}^{N_{\text{bonds}}} \frac{k_i^b}{2} \left(r_i - r_i^0\right)^2 + \sum_{j=1}^{N_{\text{angles}}} \frac{k_j^\theta}{2} \left(\theta_j - \theta_j^0\right)^2 \\
+ \quad & \sum_{k=1}^{N_{\text{dihedrals}}} \sum_{n=1}^{4} \frac{V_n}{2} \left[1 + \cos(n\phi^k - \phi_n^k)\right].
\end{aligned}
\tag{3}
$$

5. Page 4, Figure 1: The authors should label the different hydrogen bonds in the figure to facilitate cross-referencing with the d[O...H] lines in Table 1.

   **We have numbered the hydrogen bonds for dimer structures a-d in Fig. 1 and added the corresponding hydrogen bond number to the hydrogen bond distance values in the Tab. 1.**

6. Page 4, Table 1: It is curious that the authors benchmark the two force fields against a 2012 paper while more recent and more rigorous computational results should be available. The authors should reference other high quality works on the sulfuric acid dimer and justify their choice to use the 2012 paper as a benchmark.

   **The paper by Temelso et al. (2012) is to our knowledge the only reference that contains detailed information on potential/electronic energies and hydrogen bond geometries for different conformers of the $H_2SO_4$ dimer, as well as the binding free energy. For the binding free energies, we found reasonable agreement between the study by Temelso et al. (2012) and more recent work by Elm et al. (2016) and Myllys et al. (2017), which we do mention in the manuscript. Note that Temelso et al. have obtained the binding free energy from Boltzmann-averaging over the four minimum energy dimer structures, while in the newer references only the global minimum energy structure has been considered, this detail has also been added to the paragraph.**

7. Page 4, Table 1: It is curious why the authors use eV units for their $\Delta\Delta E$ values while the most commonly used unit is kcal mol$^{-1}$.

   **In computational physics and chemistry, commonly used units of energy are eV, kJ/mol, kcal/mol, or $k_B T$. The unit used in the LAMMPS simulation in/output was eV, which is why this was the most natural choice for the manuscript. Since none of the important quantities we report, such as the collision rate coefficients, or enhancement factors, have the unit of energy, we think this should not be a major concern. However, to accomodate all audiences, we have added the conversion factors at the bottom of Table 1: "Energy unit conversion: 1 eV $\approx$ 96.49 kJ$\cdot$mol$^{-1}$ $\approx$ 23.06 kcal$\cdot$mol$^{-1}$ $\approx$ 38.68$k_B T$ at $T = 300$ K."**

8. Page 8, line 16: "diameters of 49-127nm" seems incorrect. Perhaps the units are wrong.

   **The values cited in our manuscript indeed correspond to the values given in the paper by Chan and Mozurkewich (2001), both in the abstract and in Fig. 5.**

[revised manuscript text omitted]